# Depression and Autoimmune Hypothyroidism—Their Relationship and the Effects of Treating Psychiatric and Thyroid Disorders on Changes in Clinical and Biochemical Parameters Including BDNF and Other Cytokines—A Systematic Review

**DOI:** 10.3390/ph15040391

**Published:** 2022-03-24

**Authors:** Zofia Kotkowska, Dominik Strzelecki

**Affiliations:** Department of Affective and Psychotic Disorders, Central Teaching Hospital, Medical University of Łódź, ul. Czechosłowacka 8/10, 92-216 Łódź, Poland; zofia.suliga@interia.pl

**Keywords:** depression, chronic autoimmune hypothyroidism, BDNF, systematic review

## Abstract

Various autoimmune diseases, including autoimmune hypothyroidism (AHT), are associated with a higher risk of developing mood disorders throughout life. Depression is accompanied by the changes in the levels of inflammatory and trophic factors, including interleukins (IL-1beta, IL-2, IL-6), interferon alpha (IFN-alpha), tumor necrosis factor alpha (TNF-alpha), C-reactive protein (CRP), and brain derived neurotrophic factor (BDNF). Disclosure of the relationship between the coexistence of depression and AHT indicates that the pathomechanism of depression may be related to the changes in the immune system, it is also possible that both conditions may be caused by the same immune processes. The above hypothesis is indirectly supported by the observations that the treatment with both antidepressants and levothyroxine leads to a decrease in the levels of proinflammatory cytokines with an increase in BDNF concentrations, simultaneously correlating with an improvement in the clinical parameters. However, so far there are no long-term studies determining the causal relationship between depression, thyroid autoantibodies, and cytokine profile, which could bring us closer to understanding the interrelationships between them and facilitate the use of an adequate pharmacotherapy, not necessarily psychiatric. We consider the above issues to be insufficiently investigated but of great importance. This article is an overview of the available literature as well as an introduction to our research project.

## 1. Introduction

The relationship between thyroid function and depression has been known for a long time [1,2,3]. It was first described in 1825 by Parry, who noted an increased number of “nerve strokes” in thyroid disease. Seagull in 1873 showed a link between myxedema and psychosis, which was confirmed in 1888 by the Committee of the Clinical Society. In 1949 Asher introduced the term “myxedema madness” to describe mental state changes in patients with hypothyroidism [4]. It is currently well known that thyroid dysfunction can significantly affect the mental state, including emotions and cognitive functions. Both the excess and the deficiency of thyroid hormones can cause mood disorders, including depressive disorders, which can be usually resolved with an appropriate treatment of dysthyreosis. Depression may, in turn, be accompanied by various degrees of thyroid dysfunction. An overt hypothyroidism is present in 1–4% of patients with affective disorders, while subclinical hypothyroidism occurs in 4–40% of this population. According to Boswell, the frequency of depressive symptoms in patients with hypothyroidism reaches 50% [5], while depression of significant clinical severity occurs in over 40% of people suffering from hypothyroidism [6]. What is practically very important, and what prompted us to undertake the research, is that the depressive states associated with broadly understood thyroid dysfunction are usually at least partially resistant to antidepressant treatment, with the causal treatment being the key here. Thyroid hormones are also usually recommended as an adjunctive therapy in the treatment of depression in treatment algorithms [7,8]. The interdependence of thyroid disorders and depression, as well as their coexistence, is of great clinical importance, but knowledge about it is very limited. In the article, we want to summarize the current knowledge and raise topics related to the issue and which may indicate new directions of research.

### Autoimmune Hypothyroidism (AHT) and Depression

AHT is a progressive disease of the thyroid gland. Dense lymphocytic infiltration covering the gland is involved in the pathogenesis of this type of chronic thyroiditis [9]. Activated B cells produce antibodies against several major thyroid antigens. AHT is characterized by a combination of typical clinical features (Table 1) [10], increased serum thyroid-stimulating hormone (TSH) levels; decreased free thyroxine (fT4) levels; the presence of antibodies to thyroid antigens; and decreased ultrasonographic echogenicity of the thyroid parenchyma [11].

AHT is the most common organ autoimmune disorder, with an estimated prevalence of approximately 2%, women suffer more frequently. Thyroid peroxidase (TPO) is the main autoantigen, and antibodies to TPO (Anti-TPO) are present in almost all AHT patients and may precede onset of the clinical phase by up to several years. Subclinical AHT (with the presence of Anti-TPO, elevated TSH and normal fT4 levels) is more frequent and affects approximately 9% of the general population [11].

Depression, including severe depression, is more common in patients with euthyroid chronic AHT compared to subjects without this condition [9]. Frequent simultaneous presence of depressive and anxiety disorders in patients with Hashimoto’s disease (over 90% of AHT) in the euthyroid stage has also been confirmed [12]. A large Danish epidemiological study has shown that various autoimmune diseases, including AHT, are associated with a higher occurrence of mood disorders throughout life [13]. It has been shown that it is not only the decreased level of thyroid hormones that determines the occurrence of mood disorders. We also know that even in patients with normal thyroid function but with elevated Anti-TPO antibodies, their presence correlates with a higher risk of anxiety and mood disorders [14]. An increased level of anti-thyroid antibodies has been documented in 20% of patients with depression, while the incidence in the general population ranges from 5% to 10% [15,16]. In patients with bipolar disorder, regardless of the use of lithium (having “anti-thyroid” properties), the presence of Anti-TPO and AHT is also more frequent [17,18]. Also, the offspring of people with bipolar disorder have a higher incidence of Anti-TPO, even if they do not have mental disorders [19,20]. There are also data showing an increased level of Anti-TPO and antibodies blocking the TSH receptor in patients with depressive disorders without thyroid dysfunction, but who were resistant to antidepressant treatment [21]. Several hypotheses are proposed to explain the above observations. A decrease in the secretion of thyroid hormones is interpreted as a result of the dysregulation of the TSH secretion circadian rhythm (physiologically TSH levels are higher at night). This suggests that some depressed patients may have central hypothyroidism (pituitary and hypothalamus dependent, secondary, and tertiary, respectively). Moreover, sleep deprivation used in the treatment of depression leads to the restoration of the nocturnal rise in TSH levels and, consequently, an increase in free triiodothyronine (fT3) and fT4 levels. According to another hypothesis, autoimmune thyroid diseases are associated with the hypothalamic–pituitary–adrenal axis and are mediated by changes in the concentration of pro-inflammatory and anti-inflammatory cytokines. TNF-alpha (tumor necrosis factor) and interleukins IL-1 and IL-6 (all pro-inflammatory agents) increase the release of CRH (corticoliberin) and AVP (arginine vasopressin) leading to increase of glucocorticosteroid (GC) secretion from the adrenal cortex. Chronically high GC levels induce receptor resistance to GC, subsequently causing further increase in their secretion and disturbances in the functioning of the hypothalamic–pituitary–adrenal axis, increasing the susceptibility to autoimmune processes and depression [22,23]. The next hypothesis connects the frequent postpartum recurrence of depression with the end of the immune tolerance present in pregnancy [24,25]. Unfortunately, currently there are no studies showing a relationship between postpartum thyroiditis and postpartum depression and convincingly explaining the etiopathogenetic relationships but indicate the existence of common elements of the immune pathogenesis of both AHT and mood disorders.

The results of the summary on depression, anxiety, and AHT indicate that, in the United States, the prevalence of AHT is 4–13%, AHT more often affects women, and its occurrence increases with age, reaching 20% in the group of older women [26]. Prevalence of depression reached 6.6%, and anxiety disorders 18.1% in this study. It was also found that depression occurs in 16.8% of patients suffering from AHT, while the criteria for anxiety disorders were met by 35.7% of patients. The recently published meta-analysis of the comorbidity of hypothyroidism and depression did not show a statistically significant relationship between the autoimmunity of the thyroid gland and the incidence of depressive disorders [27]. The study of the relationship between morphological changes in the central nervous system in AHT patients with hypothyroidism using positron emission tomography (PET) also did not show any direct relationship between specific changes caused by AHT and the development of depressive disorders [2,28].

## 2. Methods

### 2.1. Review Question

Patients with autoimmune hypothyroidism have a potentially higher risk of depression. While coexistence of these health problems is of great clinical importance, the knowledge linking these issues is very limited. The purpose of this article is summarizing the current knowledge, and raising topics related to the issue which may indicate new directions of research. All data included in the review should meet high methodological requirements.

### 2.2. Searches

The literature search was completed on 8 March 2022. We searched five databases: EMBASE; PubMed; Cochrane Library; Scopus; and Web of Science. The search was restricted to peer-reviewed publications of original research written in English and German from 1990 to 2022, both years included. Both authors checked all the databases and collected the data separately. The inclusion of the study into the list was preceded by a discussion and the consent of both researchers.

### 2.3. Search Strategy

An example of a search strategy for PubMed database: “autoimmune hypothyroidism” and “depression”; “autoimmune hypothyroidism” and “depression” and “BDNF”; “autoimmune hypothyroidism” and “depression” and “interleukin”, autoimmune hypothyroidism” and “depression” and “gene”; autoimmune hypothyroidism” and “depression” and “treatment”.

### 2.4. Condition or Domain Being Studied

All aspects of autoimmune hypothyroidism (AHT) and depression comorbidity.

### 2.5. Participants/Population

For the purposes of this review, we included all studies that studied various aspects of depression in a population of patients with autoimmune disorders of the thyroid gland. From the scientific perspective, we were most interested in the works on autoimmune hypothyroidism (AHT), but already while writing the protocol, we noted that some literature did not use the term AHT, however, important data for us were described in patients with Hashimoto disease (at least 90% of AHT). For obvious reasons, we decided to include both diseases, remembering to follow the terminology used by the authors in the description of the results. Due to the fact that the literature in this area was limited, we tried to include all original studies on the subject. Our search included available data on changes in clinical and biochemical parameters of coexisting AHT and depression, and the effect of treatment of mental disorders and thyroid gland on changes in these parameters.

### 2.6. Intervention(s), Exposure(s)

We described all relevant data from studies focused on the relationship of autoimmune hypothyroidism (including Hashimoto disease), thyroid parameters (TSH, triiodothyronine, thyroxine, antibodies including anti-TPO, ultrasonography findings) with depression (symptomatology, drug effects incl. antidepressants); morphological; functional changes in imaging techniques (e.g., magnetic resonance imaging); differences in gene polymorphisms and gene activity; levels of interleukins; BDNF; and other relevant proteins.

### 2.7. Comparator(s)/Control

Majority of the results included in the review were compared to the control group or between groups.

### 2.8. Types of Study to Be Included

We accepted all studies that documented methodologically correct co-occurrence of disorders, which included patients with depression and hypothyroidism in accordance with the ICD or DSM terminology.

### 2.9. Main Outcomes

Correlations between parameters of autoimmune hypothyroidism as TSH; triiodothyronine; thyroxine; antibodies (including anti-TPO) levels; ultrasonography findings and parameters linked to depression (severity of symptoms, effects of treatment, incl. antidepressants and thyroid hormones) were our main outcomes. We also planned to identify and analyze morphological and functional changes in brain imaging techniques (e.g., magnetic resonance imaging); differences in gene polymorphisms and gene activity; levels of BDNF; interleukins; and other relevant proteins in patients with AHT and depression.

### 2.10. Measures of Effect

Our systematic review discusses the effect measure(s) for the main outcome(s) e.g., relative risks, odds ratios, risk difference, and/or number needed to treat if possible.

### 2.11. Data Extraction

We extracted data from the included studies for simultaneous assessment of study quality and summary of the evidence. Extracted information that can be used in a tabular form includes author; year; sample size; characteristic of participants; results and conclusions; and reference. Both authors independently extracted the data, any incongruities were resolved through discussion.

### 2.12. Risk of Bias (Quality) Assessment

We assessed quality of studies to review.

### 2.13. Certainty Assessment

All articles cited and discussed were published in peer-reviewed journals.

### 2.14. Strategy for Data Synthesis

Due to the nature of the research described, descriptive analysis was the main method used in our work.

### 2.15. Registration

PROSPERO ID 314540.

Figure 1 shows the PRISMA flow chart for screening and selection of studies researching the comorbidity of AHT and depression.

## 3. Results and Discussion

We present summarized characteristics of all included studies covering population with a simultaneous diagnosis of depression and autoimmune hypothyroidism in Table 2.

### 3.1. Inflammatory Processes, Cytokines and Growth Factors

Common etiological basis for both thyroid autoimmunity and mood disorders are also seen in similar changes of growth and differentiation in the hematopoietic and neuronal system cells and similar changes in the cytokine profile [10]. Numerous studies indicate that depression activates the inflammatory response system through increased production of well-known pro-inflammatory cytokines such as IL-1beta, IL-2, IL-6, IFN-alpha (interferon alpha), TNF-alpha, and their receptors (IL-6R, IL-1RA) [53]. It was also observed that patients with Hashimoto’s disease (HD) had higher levels of IL-17 and IL-23 compared to the control group without thyroiditis and the highest levels were obtained in HD patients in euthyrosis. TSH was negatively correlated and fT4 was positively correlated with IL-17 and IL-23 levels, whereas there was no correlation between thyroid volume calculated from ultrasonography and levels of IL-17 and IL-23. There was also a strong association between IL-17 expression and stromal fibrosis in thyroid epithelial cells, suggesting that the proinflammatory effects of IL-17 are directing thyroid tissue development towards fibrosis specific for Hashimoto’s disease that distinguishes this disease from other benign thyroid disorders [54].

The pathophysiological effects of thyroid autoantibodies in depression may depend on cytokines. Th17 lymphocytes and their primary cytokine, interleukin 17, play an important role in autoimmune diseases—IL-17 is a key signaling molecule that induces the release of pro-inflammatory cytokines and chemokines. However, to date, little is known about whether IL-17 is related to depression. One study [55] showed that IL-17 levels did not differ between controls and depressed individuals. IL-17 levels were not associated with either antithyroid antibody levels or depression severity scale. IL-17 was also not associated with age, thyroid hormone levels, or other thyroid autoantibody levels.

As we mentioned, stress induces production of pro-inflammatory cytokines leading to neuroendocrine and neurotransmitter changes resembling symptoms of depression. IFN-alpha immunotherapy (e.g., used previously in hepatitis C) often causes depressive symptoms and autoimmune thyroid disorders with the appearance of anti-TPO and anti-thyroglobulin (anti-TG) antibodies [56]. The presence of acute phase proteins and cytokines may be associated with inflammation within the brain. Peripherally produced cytokines can cross the blood-brain barrier [57,58]. After crossing the barrier they are able to participate in stress response modulation and regulation of neurogenesis [56].

Cytokines act on the brain in two successive stages: the first is triggered by activation of primary afferent neurons innervating the area of the body where the inflammatory response is taking place. The second involves the slow diffusion of cytokines from the periventricular organs and choroid plexus to target brain parts, such as the amygdala complex. Although disease behavior is a normal host response to pathogens that are recognized by the innate immune system, there is evidence that the mechanisms that contribute to the development of disease behavior may play a role in the pathophysiology of depression [59]. Mood can be attenuated also by the influence of cytokines on neurogenesis within hippocampal neurons, which is believed to be a key mechanism in the pathophysiology of depression and its treatment [60,61,62]. The weakening of neurogenesis may, over time, contribute to the reduction of the gray matter volume in the hippocampus, often observed in depression [63]. It is still unclear whether the presence of acute phase proteins may be the cause, consequence or only accompanying the depression. Studies of changes in cytokines and other inflammatory parameters in the population with the concomitant occurrence of AHT with depression (also during pharmacological interventions) are very limited.

Similarly, the direct pathophysiological relevance of antithyroid antibodies to neuropsychiatric disorders is not established yet. In one study, anti-TPO antibodies were shown to bind to cerebellar astrocytes. Another study suggested that cerebral vascular smooth muscle cells may be target structures for anti-TG antibodies. However, replication of these findings is still pending, and their relevance to the pathophysiology of depressive syndromes is unclear. Therefore, most authors consider anti-thyroid antibodies only as an epiphenomenon indicating an autoimmune predisposition. The occurrence of antibodies in healthy individuals and the independence of serum concentrations from the clinical symptoms’ expression in patients with Hashimoto’s encephalopathy indicates no direct pathophysiological significance of these antibodies [49]. On the other hand, the presence of anti-TPO antibodies during pregnancy and in the weeks after childbirth increases the risk of developing post-partum depression [50] and increases the risk of anxiety in the postpartum period [64].

A study conducted in patients with hypothyroidism after thyroidectomy due to cancer showed increased levels of the pro-inflammatory cytokines IL-6; IL-10; IL-17; TNF-alpha; and C-reactive protein (CRP). Levothyroxine therapy (used in hypothyroidism) resulted in a decrease in the level of these cytokines, but their levels were still higher than in healthy subjects [65]. In patients treated with levothyroxine, a simultaneous reduction in the level of pro-inflammatory cytokines and an increase in the levels of anti-inflammatory cytokines has been also demonstrated [66]. Interestingly, primary AHT is characterized by increased values of pro-inflammatory cytokines such as IL-2; IL-6; IL-15; TNF-alpha; and CRP [7,54,66,67]. During treatment with levothyroxine, a significant decrease in the levels of IL-1; IL-2; IL-6; IL-12; IFN-gamma; TNF-alpha; and a significant increase in IL-10 (anti-inflammatory cytokine) was observed [7,66]. In the Polish study the incidence of mild and moderate depression among patients with hypothyroidism was initially 57%. After six months of levothyroxine therapy and achieving euthyroidism, nearly half of the group (42%) remitted depressive symptoms [7].

Use of selective serotonin reuptake inhibitors (SSRI), our most frequently used group of antidepressants, reduces the promoting-depression effect of pro-inflammatory cytokines. Fluoxetine has been shown to reduce the expression of IL-1beta, IL-6, and TNF-alpha, but, interestingly, subsequently increasing the level of IL-10 [68]. Desipramine (a tricyclic antidepressant, (TCA)) reduces the levels of TNF-alpha in the hippocampus and brainstem [69], and its clinical efficacy has been associated with its ability to alter the sensitivity of noradrenergic neurons to TNF-alpha [70]. In conclusion, studies evaluating the effects of SSRIs and other antidepressive drugs indicate that inflammatory factors contribute to the pathogenesis of depression, and that various antidepressants have ability to reduce the release of pro-inflammatory cytokines, such as e.g., IL-1beta.

The results of the few published studies indicate that various antidepressants affect the level of thyroid hormones differently in patients with depression, which is probably due to differences in mechanisms of action between the drugs [71]. The results of a study evaluating the effects of reboxetine, venlafaxine and sertraline on TSH and thyroid hormones (T4—thyroxine and fT4) levels before and after treatment in severely depressed patients showed large discrepancies between drugs, although improvement in depressive symptoms occurred in all groups. A decrease in TSH levels and an increase in fT4 and T4 was observed in patients treated with reboxetine, no changes in hormone levels were observed in those treated with venlafaxine, while patients taking sertraline had an increase in TSH levels, a decrease in fT4, and T4 levels [72]. The effect of SSRIs on thyroid function was summarized by a meta-analysis that included clinical studies that measured levels of thyroid parameters (TSH, T4, fT4, or fT3) before and after treatment with SSRIs (as a group). It was shown that after treatment with these drugs, patients had lower levels of T4, fT4 and fT3 with no change in TSH values [73]. So far, the mechanism of inducing hypothyroidism during SSRI treatment has not been established. One hypothesis is that SSRIs stimulate the activity of the type 2 iodothyronine deiodinase enzyme, which converts T4 to T3 (trioiodothyronine) in various tissues, including the brain [74]. Also, the relief of depressive symptoms causes biological effects that may modulate the thyroid axis. It has been documented that there is an association of the effect of venlafaxine treatment with polymorphisms in the NR3C2 gene and elevated TSH levels [75] in a mechanism to restore normal function of the hypothalamic–pituitary–adrenal axis through regulation of the mineralocorticoid receptor (MR) and the glucocorticoid receptor (GCR), encoded by the NR3C2 and NR3C1 genes. The NR3C2 gene provides instructions for the formation of a protein called the mineralocorticoid receptor. This protein plays the important role in regulating the amount of sodium in the body. Sodium co-regulates blood pressure control and fluid balance. Mineralocorticoids attach (bind) and switch on (activate) the mineralocorticoid receptor. Aldosterone is one of the mineralocorticoids that activates the mineralocorticoid receptor. Activated mineralocorticoid receptor acts as a transcription factor, a protein that binds to specific DNA regions and helps control the activity (transcription) of particular genes. The mineralocorticoid receptor regulates specialized proteins in the cell membrane that control the transport of sodium or potassium into cells. In response to signals that sodium levels in the body are low, the mineralocorticoid receptor increases the number and activity of these proteins in the cell membrane, especially in selected kidney cells. One of these proteins transports sodium into the cell, while another protein simultaneously transports sodium out of the cell and potassium into the cell. These proteins help keep sodium in the body (reabsorption) and remove potassium from the body (secretion) [76].

Studies on the effect of thyroid hormones on the speed of action of antidepressants indicate that accelerating the response to antidepressants is not possible for all classes of antidepressants. T3 accelerates responses to TCAs (tricyclic antidepressants) but does not have the same effect when used with SSRIs. In a meta-analysis of four RCTs of controlled depression patients, there was no evidence of a faster onset of response when T3 was added to SSRIs [77]. The reasons for a discrepancy between the effects of T3 on TCA versus SSRIs remain unknown [78]. However, we do not have data on the effect of treatment with antidepressants on the biochemical parameters of the thyroid gland, including the levels of anti-thyroid antibodies in patients with AHT. Figure 2 summarizes the interrelationships between treatment, biochemical parameters and clinical outcome.

It is worth noting that preliminary data suggest that anti-inflammatory drugs may be useful in mood disorders—it has been reported that in patients treated with rofecoxib and celecoxib, the depressive symptoms’ improvement was more pronounced than in the group not treated with these drugs [79,80]. This effect, at least partially, may be related to the analgesic effect of the COX-2 inhibitors, but studies on rats showed that use of rofecoxib leads to a serotonin increase in the frontal, parietal, and temporal cortexes, which may indicate their other-than-analgesic mechanism of action [79]. So far, no studies have been conducted to assess the effect of anti-inflammatory drugs on the biochemical parameters of thyroid function in AHT. Even if not all depressive states have an inflammatory etiology, according to the available data there is possibly a separate subtype of depression of inflammatory origin, or a manifestation of an inflammatory process is able to clinically mimic depression. Biological anti-inflammatory drugs, including anti-TNF-alpha antibodies such as e.g., infliximab, adalimumab, and guselkumab potentially may be useful to treat those types of depression. Those particles have an ability to reduce the severity of inflammation (e.g., in rheumatic diseases, psoriasis, and Crohn’s disease) very effectively. However, the conclusions from limited depression trials are ambiguous, although patients with high initial levels of pro-inflammatory cytokines may have more benefit with this treatment [81,82,83]. Tocilizumab (IL-6 receptor blocker) studies show its potential to decrease the levels of depressive symptoms in patients with rheumatoid arthritis [84,85], but in patients with haematological problems the results were completely different—depressive symptoms were exacerbated with this drug [86]. Interestingly, hallucinogens (increasingly studied in depression) and bupropion have an ability to block TNF-alpha or decrease its production [87].

Scarce data from studies concerning the thyroid gland function during use of TNF-alpha antibodies showed mostly neutral results (no changes in TSH, T3, anti-TPO and anti-TG antibodies levels, decrease in fT4 levels) [88]. We need to underline here that depression (also with increased suicidal risk) and thyroiditis (new onset or its exacerbation) are among common side effects of TNF-alpha blockers.

Studies involving relatives of patients with AHT force even deeper reflections about the common origin of both AHT and depression. Euthyroid women being relatives of AHT patients have abnormal serum levels of hematopoietic and neural growth and differentiation factors important in the etiology of depression—BDNF (Brain-derived neurotrophic factor); IGFBP-2 (insulin-like growth factor binding protein); EGF (epidermal growth factor); and SCF (stem cell factor) [11]. Serum levels of the other growth factor TGF-beta1 (a polypeptide member of the transforming growth factor beta superfamily of cytokines) are lowered in patients with HD [89].

It is also worth noting the increasing research on the impact of gut microflora and translocation of bacteria from the gut to the blood in patients with depressive disorders. The relationship between gut barrier dysfunction, bacterial translocation, and the immune system in depressed patients and the expression of activated monocytes in depressed patients has been established. Treg lymphocytes (which play a role in regulating the activation and effector activity of innate and adaptive immune response cells, as well as in controlling and suppressing the immune response against their own antigens) play an important role in this mechanism. There is an increased proportion of Treg cells in depressed patients. Moreover, there is a significant association between the gut microbiota and Treg. The reduction of Treg expansion may suppress the intestinal migration of these cells. Reduction of Treg of systemic origin may compromise the total number of intestinal Treg. Reduction of Treg in the intestinal mucosa may promote intestinal barrier damage and increased bacterial translocation found in some depressed patients. Furthermore, the presence of depression is associated with increased intestinal permeability or ‘leaky gut’ and increased bacterial translocation. There is also evidence that the microbiota affects the immune system and vice versa [9].

Depressed patients had significantly elevated levels of circulating intestinal fatty acid-binding protein, a marker of intestinal barrier function in peripheral blood (I-FABP) and lipopolysaccharide-binding protein (LBP). In addition, depressed patients with high LBP levels showed a significant reduction in circulating Treg compared to depressed patients with normal LBP. The balance of effector lymphoid cells and Treg cells may have a profound effect on how the intestinal mucosa responds to stressors that cause damage. Although there is no direct evidence for causality, this study supports the existence of a so-called gut–brain axis, linking gastrointestinal and immune system functions to the emotional and cognitive areas of the brain [90].

### 3.2. Brain-Derived Neurotrophic Factor (BDNF) and Depression, AHT and Gender

BDNF needs a broader description as a crucial element in the current understanding of the development and dynamics of depressive symptomatology. This particle belongs to the group of neurotrophins, proteins synthesized in the cells of the central and peripheral nervous system and involved in the development, function, and protection of nerve cells. It regulates many processes in our body, including the development and growth of neurons, inhibiting apoptosis, promoting neurogenesis, neuroregeneration, and stimulating the formation of dendritic connections. BDNF participates in the regulation of neuronal plasticity related to learning and memory processes, influencing the process of synaptic long-term potentiation and long-term depression in the hippocampus. It also influences the development of serotonergic, dopaminergic, noradrenergic, and cholinergic neurons. Dopaminergic neurons of the substantia nigra and striatum have been found to be the main source of BDNF secretion. BNDF easily crosses the blood-brain barrier [91].

Preclinical studies show that stress reduces BDNF expression in the rat hippocampus, while a single two-sided direct infusion of BDNF into the rat hippocampus has an antidepressant effect in animal models of depression [92]. In humans, BDNF plays a significant role in the pathophysiology of mental disorders, especially depression, where its key role is undisputable. Patients with severe depressive symptoms show lower levels of BDNF compared to controls. Moreover, BDNF levels are correlated with the reduction of the hippocampal volume [93]. We also know that BDNF expression is lower in the prefrontal cortex and hippocampus of people who died by suicide compared to the control group matched in terms of sex and age [94]. The use of sertraline, escitalopram, and venlafaxine in the study by Matrisciano et al. resulted in significant clinical improvement despite varying effects on the peripheral BDNF levels. A relationship was also found between an increase in BDNF serum levels and an improvement in the Hamilton Depression Rating Scale, thus indicating that a higher BDNF level in the blood serum corresponds to recovery [93]. It has been shown that BDNF levels were directly related to antidepressant responses, and people who responded well to treatment (>50% improvement in the scores of depression severity scales) had higher BDNF levels before treatment than non-responders, indicating that BDNF can be also a potential predictor of the antidepressant response [95]. It was also confirmed that the concentration of BDNF and its changes are not rigidly correlated with improvement in depression, but that the level of BDNF generally increases during antidepressant treatment. Studies on rapid-acting antidepressant—ketamine—having different mechanisms of action than SSRIs and TCAs, showed that the rapid antidepressant response after its administration is mediated by an increase in BDNF levels [96].

Research directly linking BDNF to thyroid disorders is scarce at present. Rats with early-onset hypothyroidism exhibited lower levels of BDNF in the brain [97]. Preclinical studies conducted by Hung indicate that thyroxine protects against white matter damage by increasing the level of BDNF [98]. In clinical trials, higher TSH levels were associated with a more discrete increase in serum BDNF levels in depressed patients during antidepressant therapy [99], and lower baseline TSH levels correlated with greater improvements after fluoxetine and sertraline [100]. It has also been shown that observed clinical results were better among patients who had lower TSH levels during sertraline and triiodothyronine therapy [98]. On the other hand, the only study evaluating the effect of thyroid hormones on changes in BDNF in serum, plasma, and platelets over the 3-month period of treatment with antidepressants in patients without thyroid disease showed that higher TSH levels correlated with a lower increase in serum levels of BDNF during antidepressant treatment. It was additionally indicated that with an increase in TSH, BDNF concentrations decreased within the observation period [99]. In patients participating in this study, no such relationship was found between the levels of triiodothyronine (T3); thyroxine (T4); corticotropic hormone (ACTH); cortisol; prolactin (PRL); luteinizing hormone (LH); follicle stimulating hormone (FSH); estrogen; and progesterone. A study evaluating similar relationships in patients with hypothyroidism and subclinical hypothyroidism has not been conducted so far.

In conclusion, it can be hypothesized that thyroid hormones may affect the response to antidepressant therapy through its influence on BDNF, but so far there are not sufficient data to confirm this hypothesis unequivocally. 

Due to the epidemiology of depression and AHT, it is worth emphasizing that the relationship between the level of BDNF and depression may depend on gender. It has been shown that reduced BDNF values are more pronounced in women with depression, and long-term antidepressant use selectively raises its concentration in women. Therefore, it is possible that the more frequent occurrence of depression in women may be related to this mechanism and interactions with female sex hormones, especially since it has been shown that BDNF expression can be reduced by higher levels of estradiol in animal model [101].

It is still an open question: what is the cause of the increased incidence of depression in patients with AHT? Is it the influence of hormonal disorders or changes in cytokine profile, BDNF levels, or antithyroid antibodies accompanying AHT? 

Trying to answer these questions we analyzed another autoimmune disease that is not accompanied by hormonal changes. In rheumatoid arthritis prevalence of depression is significantly higher than in the general population and is estimated at 15–34%. Studies investigating the associations of disease intensity, levels of pro-inflammatory cytokines, and neurotrophic factors with depression in patients with RA found that pro-inflammatory cytokine levels were not associated with depression, and only BDNF levels were associated with depression. Considering the above, we conclude that the reason for the higher prevalence of depression in AHT patients is endocrine disruption rather than increased levels of pro-inflammatory cytokines.

Figure 3 summarize the current knowledge on postulated pathogenetic mechanisms and relationships of depression, thyroid dysfunction, and inflammation.

## 4. Conclusions

People with a higher risk of developing mood disorders, including depressive disorders, are at the same time more likely to develop autoimmune thyroid disease and vice versa, which may indicate common pathogenetic roots. An abnormal profile of haemopoietic and neuronal growth factors, including BDNF, is observed in patients with mood disorders as well as in those at risk of developing AHT. Similar observations apply to the cytokine profile in patients with both diseases, in whom we observe an increase in the concentrations of pro-inflammatory interleukins, e.g., IL-1beta, IL-2, IL-6, and TNF-alpha. From the clinical perspective, the conclusion that seems to be of particular importance is that in euthyroid patients with autoimmune thyroiditis and elevated levels of anti-thyroid antibodies, the incidence and severity of depression is significantly higher [8]. Treatment with both antidepressants and levothyroxine leads to a decrease in the level of pro-inflammatory cytokines, an increase in the level of BDNF, correlating with an improvement in clinical parameters of depression. Hence one of the hypotheses that depression may belong to the spectrum of inflammatory and degenerative disorders [102].

For the above reasons, patients with depression and anxiety disorders should be tested for autoimmune hypothyroidism, and patients with AHT should be screened for psychiatric symptoms [26]. However, it is necessary to broaden the scope and in-depth research, also conduct long-term studies (as our team is planning now) to determine the causal relationship between depression, thyroid autoantibodies, and cytokine levels, which would help us bring us closer to understanding the interrelationships between them and facilitate the use of adequate pharmacotherapy, not necessarily psychiatric. At the same time, it would be possible to identify groups at a higher risk of the occurrence of both frequent and very burdensome diseases, from a psychiatric perspective to help prevent the development of full-blown depression, and finally to predict the therapeutic response to treatment (drugs or groups of drugs). 

Finally, we want to emphasize that our article has limitations: the results of the summary do not provide grounds for drawing clear conclusions due mainly to the fragmentary nature of the available data.

## Figures and Tables

**Figure 1 pharmaceuticals-15-00391-f001:**
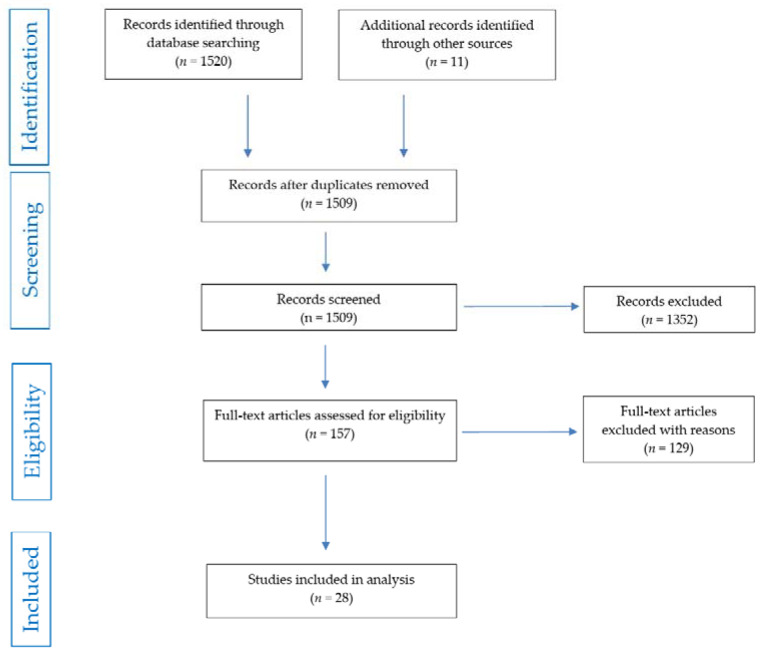
PRISMA flow chart of studies selection.

**Figure 2 pharmaceuticals-15-00391-f002:**
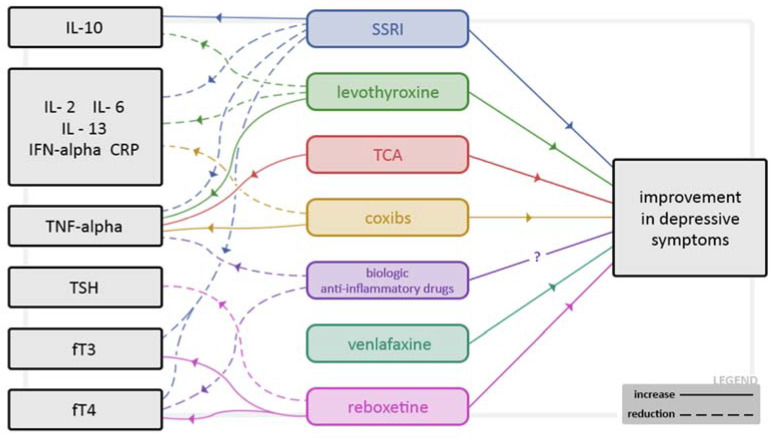
Potential mechanisms of relation between depressive symptoms and selected drugs and their influence on the values of selected hormonal and inflammatory parameters.

**Figure 3 pharmaceuticals-15-00391-f003:**
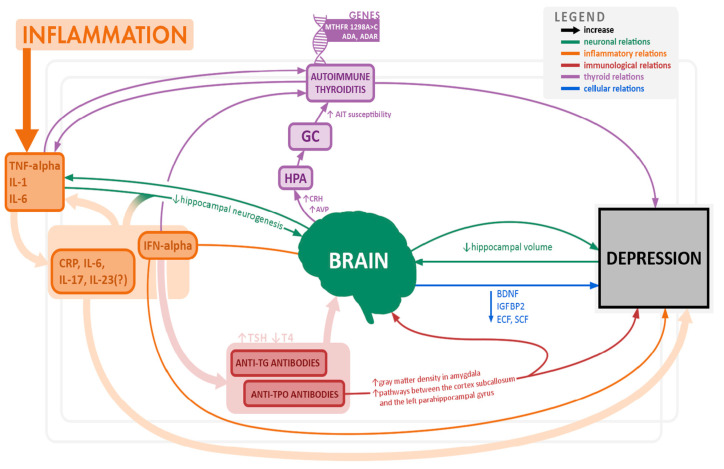
Overview of potential pathomechanisms associated with depression and thyroid dysfunction.

**Table 1 pharmaceuticals-15-00391-t001:** Autoimmune hypothyroidism—symptoms, assessment, and diagnostic findings.

Symptoms	Assessment Findings	Diagnostic Findings
Depression	Dry, coarse skin	Hyponatremia
Fatigue	Reduced body and scalp hair	Macrocytic anemia
Weight gain	Dull facial expression	Decreased memory
Constipation	Bradycardia	Hyperprolactinemia
Muscle cramps, arthralgias	Goiter	Elevated creatine kinase level
Menorrhagia	Macroglossia	Pituitary gland enlargement
Infertility	Ascites	Delayed bone age
Sexual dysfunction	Galactorrhea	Hypercholesterolemia
Cold intolerance	Slow relaxation of tendon reflexes	
Carpal tunnel syndrome	Nonpitting edema of lower extremities	
Sleep disorders	Hoarseness	

**Table 2 pharmaceuticals-15-00391-t002:** Studies investigating various aspects of depression and AHT comorbidity.

Author	Year	Sample Size	Characteristic of Participants	Results and Conclusions	Reference
Pop et al.	1998	583 women	A total of 58 women had elevated levels of anti-TPO antibodies. Age group: 47–54 years old.	This study showed that women with elevated levels of anti-TPO antibodies are more prone to developing depression, while postmenopausal age does not increase this risk.	[29]
Zetting et al.	2003	76	A total of 41 patients with autoimmune thyroiditis and 35 patients in the control group.	Brain impaired perfusion has been confirmed in patients with AHT. The presence of cerebral hypoperfusion suggests cerebral vasculitis as the most likely pathogenic cause.	[30]
Carta et al.	2004	222	In all, 16.6% of the studied patients had elevated levels of anti-TPO antibodies.	A relationship has been demonstrated between the presence of a lifetime diagnosis of depressive disorders and the level of anti-TPO antibodies.	[13]
Carta et al.	2005	190	A total of 19 patients were diagnosed with Hashimoto’s disease (HD) in euthyroidism, 19 patients with euthyroid neutral goiter, 152 people in the control group.	Patients diagnosed with HD in euthyroidism showed a higher incidence of depressive episodes throughout life.	[11]
Engum et al.	2005	30,175	In all 995 of the study group had elevated levels of anti-TPO antibodies. Age group: 40–84.	In the group with elevated levels of anti-TPO antibodies, the incidence of depression was not higher than in the general population.	[31]
Gulseren et al.	2006	160	A total of 33 patients with overt hypothyroidism, 43 patients with subclinical hypothyroidism, 51 patients with overt hyperthyroidism, 13 patients with subclinical hyperthyroidism and a healthy control group of 20 patients.	Achieving euthyroidism reduces depressive symptoms. In this study, the causes of hypothyroidism were not mentioned—the levels of TSH, fT3 and fT4 were examined.	[32]
Bunevicius et al.	2007	474	Of the 474 randomly selected primary care patients, 348 were female, 95 of them were postmenopausal, 67 had a history of endocrine disease; 68 patients had a history of mental disorders. 84 patients used psychotropic drugs, including 69 benzodiazepines; none of the patients used lithium. Six women were diagnosed with hypothyroidism and were treated with L-thyroxine.	The results of this study indicate that ultrasound-assessed thyroid autoimmunity is associated with the symptoms of mood disorders in primary care patients, especially in premenopausal women.	[33]
van der Deure et al.	2008	141	Patients with AHT treated with levothyroxine or combination of levothyroxine and liothyronine.	OATP1C1 polymorphisms are associated with fatigue and depression in patients with primary autoimmune hypothyroidism.	[34]
Schinhammer et al.	2010	96	67 patients were diagnosed with HD and 30 patients in the control group, age group 18–80.	Patients with HD have an increased risk of developing depression. This study showed that the MTHRF 1298C/C gene polymorphism in patients with HD may be associated with the etiology of depression. There seems to be an association between the MTHFR 677C/C and COMT A/A polymorphisms and Hashimoto’s disease. Gene polymorphism in the folate/homocysteine system appears to play a role in HD and associated depressive disorders.	[35]
Kirim et al.	2012	201	Patients diagnosed with HD in the euthyroid stage, age group 18–65 years.	Increased frequency and severity of symptoms of depression in patients diagnosed with Hashimoto’s disease in the euthyroid stage.	[8]
Watt et al.	2012	199	Patients with AHT.	Health-related quality of life (Qol) in patients with autoimmune hypothyroidism was related to anti-TPO antibodies level but not to thyroid function. Level of anti-TPO antibodies was significantly associated with several QoL outcomes such as depressivity, anxiety, emotional susceptibility and impaired social life. In the multivariate model, the anti-TPO antibodies levels were related to e.g., depressivity and anxiety.	[36]
Franke et al.	2013	57	36 patients diagnosed with HD and 21 patients diagnosed with neutral goiter.	The study shows increased expression of the ADA (adenosine deaminase gene) and ADAR (adenosine deaminase gene, RNA specific) genes in patients diagnosed with HD compared to patients with neutral goiter, which may explain the increased incidence of depression in patients with HD.	[37]
Giynas Ayhan et al.	2014	164	51 patients diagnosed with HD in euthyroidism, 45 patients with euthyroid neutral goiter, 68 patients in the control group.	Depressive disorders are more common in euthyroid patients diagnosed with HD, suggesting that they may be associated not only with abnormal levels of thyroid hormones, but also with the process of autoimmunity.	[38]
Medici et al.	2014	7983	1503 people had the level of anti-TPO tested, the study included Caucasian people aged 55+ (people with dementia were excluded).	Older people with low normal TSH levels have more comorbid depressive symptoms and a significantly increased risk of developing a depressive syndrome later in life. Low TSH levels are an important risk factor for the development of depression in the elderly. Autoimmunity of the thyroid gland (as assessed by elevated levels of anti-TPO antibodies) was not associated with an increased risk of depression.	[39]
Demartini et al.	2014	246	A total of 123 patients with subclinical hypothyroidism, including 106 patients diagnosed with HD, 12 with non-autoimmune hypothyroidism and 5 with nodular goiter, and 123 controls without diagnosed thyroid disease. Patients diagnosed with intellectual disability and dementia were excluded.	More than twice more patients diagnosed with subclinical hypothyroidism had at least mild depression.	[40]
Quinque et al.	2015	36	18 patients treated for AHT and 18 patients in the control group. Structural and functional MRI and neuropsychological tests were performed to assess mood and cognitive function.	Properly treated patients report more depressive symptoms compared with healthy controls. Mood changes were not associated with brain structure and function in brain regions specific to depression. Higher levels of anti-TPO antibodies are associated with higher gray matter density in the right amygdala and increased connections between the cortex subcallosum and the left post-hippocampal gyrus. Duration of treatment was associated with the development of structural and functional changes in brain areas associated with depression and untreated hypothyroidism. Autoimmunity and the duration of treatment are possible factors explaining the occurrence of psychiatric symptoms in patients receiving long-term treatment for hypothyroidism.	[41]
Itterman et al.	2015	2142	The analysis included 498 patients with previously diagnosed thyroid dysfunction—247 people were taking medication, 223 had thyroid nodules, 74 were diagnosed with hyperthyroidism, and 70 with hypothyroidism—without specifying the causes of the disorders.	Untreated, diagnosed hypothyroidism is associated with a higher risk of depressive symptoms. TSH and anti-TPO levels were not significantly associated with the risk of depression.	[42]
Fjaellegaard et al.	2015	8214	The patients were divided into four groups: 1. Euthyroid patients with normal levels of anti-TPO antibodies (7015 patients), 2. Euthyroid patients with elevated levels of anti-TPO antibodies (619 patients), 3. Patients with subclinical hypothyroidism and normal levels of anti-TPO antibodies (378 patients), 4. Patients with subclinical hypothyroidism and elevated levels of anti-TPO antibodies (202 patients).	This study showed no significant differences in the incidence of depression in euthyroid patients and patients diagnosed with subclinical hypothyroidism. Euthyroid women with elevated levels of anti-TPO antibodies had statistically significantly better well-being than patients with normal levels of anti-TPO antibodies.	[43]
Van de Ven et al.	2016	906	Age group 50–70, relationship between the presence of anti-TPO antibodies, TSH and fT4 levels and the risk of depression was examined.	Presence of anti-TPO antibodies may be a marker of susceptibility to depression. Lack of correlation between thyroid function and incidence of depression.	[44]
Krysiak et al.	2016	86	Age group: women aged 20–40, 68 patients were divided into four groups: 1. Patients in euthyroidism diagnosed with HD, 2. Patients with non-autoimmune hypothyroidism, 3. Patients with autoimmune hypothyroidism, 4. Control group—18 patients.	The Beck Depression Inventory (BDI) total score was highest in group 3, and higher in groups 1 and 2 than in group 4. Anti-TPO antibody levels were directly proportional to serum TSH levels and the BDI total score and the number of patients with depressive symptoms.	[45]
Delitala et al.	2016	3138	The group included patients who were not taking thyroid medications or antidepressants. The levels of TSH, fT4 and anti-TPO antibodies were assessed.	No relationship was found between the level of anti-TPO antibodies and the occurrence of depressive symptoms. On the other hand, a U-shaped relationship was found between the level of fT4 and the occurrence of depressive symptoms in comparison with the average values of fT4—both high and low values of fT4 were associated with a greater number of depressive symptoms.	[46]
Yalcin et al.	2017	124	93 patients diagnosed with euthyroid HD for at least 3 months and 31 patients in the control group.	The level of TSH was statistically higher in patients diagnosed with HD, no differences in the level of fT4 were observed in the group of patients with HD and in the control group. In 17.3% of patients diagnosed with HD and in 4.3% of patients in the control group, depression was diagnosed. Autoimmunity itself may have an impact on the risk of depression in patients diagnosed with HD in the euthyroid stage.	[47]
Bhagwat et al.	2017	66	33 patients with autoimmune hypothyroidism and 33 patients from the control group.	In 57% of patients diagnosed with AHT, mild to moderate depression was diagnosed (MADRS > 11 points). After 6 months of treatment with tyroxin, 42% of these patients had remission of symptoms. The decrease in inflammatory markers correlated with the remission of depression.	[4]
Lee et al.	2019	1651	The study group was divided into three groups depending on the level of TSH. Anti-TPO antibodies and fT4 levels were also tested in all patients.	Depressive symptoms were observed less frequently in patients with positive anti-TPO antibodies and the highest TSH concentrations than in the group of patients with the lowest TSH concentrations. Men with the highest TSH level were less than twice as likely to develop depressive symptoms than in the group with the lowest TSH levels. In women with the highest TSH level 35% less often depressive symptoms than in the group with the lowest TSH level. Gender may play an important role in the relationship between TSH levels and depressive symptoms.	[48]
Dersch et al.	2020	100	100 patients with unipolar endogenous major depression or treatment-resistant depression, including: 25 patients with first depressive episode (6 of them patients with psychotic symptoms) and 75 patients with recurrent depression (18 of them were patients with psychotic symptoms).	This study provides evidence of intrathecal synthesis of anti-thyroid antibodies in a subset of patients with unipolar depression. This may indicate central immunization in a subset of patients diagnosed with HD.	[49]
Minaldi et al.	2020	2932	A meta-analysis of five studies (449 women with anti-TPO antibodies and 2483 women without anti-TPO antibodies).	Thyroid autoimmunity (anti-TPO antibodies -positive) during pregnancy and in the weeks after childbirth is associated with an increased risk of developing post-partum depression.	[50]
Hirtz et al.	2021	360	Adolescents (11–19 years) with at least mild depression (BDI-II score > 13) compared to a representative reference cohort without evidence of mental health impairment.	Study found a higher prevalence of thyroid peroxidase antibody positivity in depressed adolescents compared to mentally healthy participants. The prevalence of subclinical hypothyroidism was higher in depressed adolescents vs. mentally healthy participants, but no other types of thyroid dysfunction had a higher prevalence. The prevalence of subclinical hypothyroidism and of thyroid autoimmunity in depressed adolescents is increased.	[51]
Kamyshna	2022	153	Patients with various forms of thyroid pathology including AHT.	The CT genotype of the glutamate ionotropic receptor NMDA type subunit 1, GRIN1 (rs4880213) was predominant in the patients with autoimmune diseases of the thyroid gland. GRIN2B (glutamate ionotropic receptor NMDA type subunit 2B) serum levels in ELISA were probable increased in AHT by 1.58 times compared with controls, while significantly (by 3.45 times) decreased in patients with postoperative hypothyroidism. The C allele of rs4880213 was more frequent than the T allele among patients with thyroid disease. GRIN2B levels were significantly different in patients of different groups depending on thyroid pathology. Study found direct close correlation (r = 0.635) between GRIN2B and anti-TPO levels (*p* < 0.001) and direct close correlation (r = 0.527) between GRIN2B and anti-TG levels in the blood (*p* < 0.001).	[52]

Abbreviations: TPO—thyroid peroxidase; AHT—autoimmune hypothyroidism; HD—Hashimoto’s disease; TSH—thyroid-stimulating hormone; fT3—free triiodothyronine; fT4—free thyroxine; MRI—magnetic resonance imaging; MADRS—The Montgomery–Åsberg Depression Rating Scale; BDI—Beck Depression Inventory; NMDA—N-methyl-D-aspartate; GRIN2B—allelic variants of N-methyl- d-aspartate receptor 2B.

## Data Availability

Data sharing is not applicable to this article as no new data were created in this study.

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
