# Peer review of "Depression and Autoimmune Hypothyroidism—Their Relationship and the Effects of Treating Psychiatric and Thyroid Disorders on Changes in Clinical and Biochemical Parameters Including BDNF and Other Cytokines—A Systematic Review"

_pharmaceuticals, 2022, doi:10.3390/ph15040391_

Round 1

Reviewer 1 Report

A review article by Kotkowska and Strzelecki presents the association between depression and autoimmune thyroiditis and the effects of the treatments of these diseases on clinical and biochemical parameters. In general, the article is prepared in a proper manner and reads well. However, there are some issues that should be clarified before the acceptance.

Major concerns.

  1. The authors omitted the role of IL-17 which plays a crucial role in autoimmunity. They should include this cytokine in more detail e.g. Ann Acad Med Singap. 2015 Aug;44(8):284-9.
  2. Line 191. Please explain the role of the NR3C2 gene and the effects of polymorphisms within it. So that a non-specialist can understand the context.

Author Response

Dear Reviewer,
thank you for review, we change the text according to your remarks:

  1. The authors omitted the role of IL-17 which plays a crucial role in autoimmunity. They should include this cytokine in more detail.

Thank you for this suggestion. We have added information on the role and level of IL-17 in patients diagnosed with Hashimoto's thyroiditis and the correlation of IL-17 level with hormonal parameters and its impact on the process of fibrosis typical of thyroid disease. We also elucidated the role of Il-17 in the autoimmune process and its potential influence on the possibility of depression. Information was added that:

"It was also observed that patients with Hashimoto's disease (HD) had higher levels of IL-17 and IL-23 compared to the control group without thyroiditis and the highest levels were obtained in HD patients in euthyrosis. TSH was negatively correlated and fT4 was positively correlated with IL-17 and IL-23 levels, whereas there was no correlation between thyroid volume calculated from ultrasonography and levels of IL-17 and IL-23. There was also a strong association between IL-17 expression and stromal fibrosis in thyroid epithelial cells, suggesting that the proinflammatory effects of IL-17 are directing thyroid tissue development towards fibrosis specific for Hashimoto's disease that distinguishes this disease from other benign thyroid disorders [48]”.         
and

„The pathophysiological effects of thyroid autoantibodies in depression may depend on cytokines. Th17 lymphocytes and their primary cytokine, interleukin 17, play an important role in autoimmune diseases - IL-17 is a key signaling molecule that induces the release of pro-inflammatory cytokines and chemokines. However, to date, little is known about whether IL-17 is related to depression. One study [49] showed that IL-17 levels did not differ between controls and depressed individuals. IL-17 levels were not associated with either antithyroid antibody levels or depression severity scale. IL-17 was also not associated with age, thyroid hormone levels, or other thyroid autoantibody levels”.

2. Please explain the role of the NR3C2 gene and the effects of polymorphisms within it. So that a non-specialist can understand the context. Thank you for this remark. We explained the role of the NR3C2 gene in the text, taking into account its role in increasing TSH levels in patients treated with venlafaxine. The following part has been added:

„It has been documented that there is an association of the effect of venlafaxine treatment with polymorphisms in the NR3C2 gene and elevated TSH levels [68] in a mechanism to restore normal function of the hypothalamic-pituitary-adrenal axis through regulation of the mineralocorticoid receptor (MR) and the glucocorticoid receptor (GR), encoded by the NR3C2 and NR3C1 genes. The NR3C2 gene provides instructions for the formation of a protein called the mineralocorticoid receptor. This protein plays the important role in regulating the amount of sodium in the body. Sodium co-regulates blood pressure control and fluid balance. Mineralocorticoids attach (bind) and switch on (activate) the mineralocorticoid receptor. Aldosterone is one of the mineralocorticoids that activates the mineralocorticoid receptor. Activated mineralocorticoid receptor acts as a transcription factor, a protein that binds to specific DNA regions and helps control the activity (transcription) of particular genes.
The mineralocorticoid receptor regulates specialized proteins in the cell membrane that control the transport of sodium or potassium into cells. In response to signals that sodium levels in the body are low, the mineralocorticoid receptor increases the number and activity of these proteins in the cell membrane, especially in selected kidney cells. One of these proteins transports sodium into the cell, while another protein simultaneously transports sodium out of the cell and potassium into the cell. These proteins help keep sodium in the body (reabsorption) and remove potassium from the body (secretion) [69]. 

Authors

Reviewer 2 Report

This paper is a nice review adressing a rather difficult issue i.e the links between depression and autoimmune thyroiditis. The main general drawback here is that it is almost impossible to discriminate the impact of thyroid hormone imbalance from the effects of cytokines and autoimmunity. In this view, it would be more convincing if authors could extend their review to at least one "control" autoimmune pathology in which no obvious hormonal changes are observed (for e.g. rheumatoid arthritis). A sufficiently well documented paragraph on this "control" pathology would be enough

Then the authors could distinguish potential mechanisms which are specific to autoimmune thyroiditis from potential mechanisms which are common to autoimmune disorders, irrespective of the targeted organ. Such mechanisms likely include the presence of autoantibodies which may potentially target neural cells. What do we know about the expression of TPO and TG by neural cells ?

Another mechnisms, not specific to autoimmune thyroiditis nore autoimmune disorders, is peripheral inflammation, irrespective of its origin.

In this regard, the paper is missing an importnat point which is the impact of gut microbiota and bacterial translocation from gut to blood. Increased LPS levels (or LBP levels) have been reported in depressive patients (for e.g. PMID: 33488424). This should be discussed as this mechanism may trigger both depression and autoimmunity (https://www.sciencedirect.com/science/article/pii/S0092867416303981)

Another important point is that the authors never refer to the term "Sickness behavior" and the steming works from Robert Dantzer who, to my knowledge, was one of the the first researcher working on the links between peripheral inflammation and behavior. Here is an article worth to read and reference : PMID: 11259077. It explains how peripherally-expressed pro-inflammatory cytokines gain access to the brain via two routes (CVO vs peripheral axons).

Besides these remarks, I think that at least 1 explanatory scheme (2 or 3 even better) would add much value to this paper.

Author Response

Dear Reviewer,
thank you for review, we change the text according to your remarks:

1. The main general drawback here is that it is almost impossible to discriminate the impact of thyroid hormone imbalance from the effects of cytokines and autoimmunity. In this view, it would be more convincing if authors could extend their review to at least one "control" autoimmune pathology in which no obvious hormonal changes are observed (for e.g., rheumatoid arthritis). A sufficiently well documented paragraph on this "control" pathology would be enough.              
Thank you for this valuable comment. We analyzed a study evaluating the relationship of disease activity, pro-inflammatory cytokines and neutrophic factors with depression in RA patients, which showed that the incidence of depression in this group of patients, depending on the criteria, is between 15 and 34%, which is significantly higher than in the general population. It was also established that the level of pro-inflammatory cytokines was not associated with the occurrence of depression in patients with RA. On the other hand, among neutrophic factors, it was shown that only BDNF levels were associated with depression in RA patients. Considering the above, it can be concluded that the cause of the higher incidence of depression in patients with AIT is hormonal disorders, and not the increased level of pro-inflammatory cytokines. However, this study has several limitations, hence it is difficult to answer this question unequivocally. (Page 13)

2. Then the authors could distinguish potential mechanisms which are specific to autoimmune thyroiditis from potential mechanisms which are common to autoimmune disorders, irrespective of the targeted organ. Such mechanisms likely include the presence of autoantibodies which may potentially target neural cells. What do we know about the expression of TPO and TG by neural cells?

This is a good question. The direct pathophysiological relevance of anti-thyroid antibodies to neuropsychiatric disorders has not yet been established. One study showed that anti-TPO antibodies bind to cerebellar astrocytes. Other studies have suggested that cerebral vascular smooth muscle cells may be target structures for anti-TG antibodies. However, replication of these results is still ongoing and their relevance to the pathophysiology of depressive syndromes is also unclear. Therefore, most authors consider antithyroid antibodies to be merely an epiphenomenon indicative of an autoimmune predisposition. The presence of antibodies in healthy subjects and the independence of serum concentrations from the expression of clinical symptoms in patients with Hashimoto's encephalopathy also argue against the direct pathophysiological significance. (Page 9)    

3. Another mechnisms, not specific to autoimmune thyroiditis nore autoimmune disorders, is peripheral inflammation, irrespective of its origin.In this regard, the paper is missing an importnat point which is the impact of gut microbiota and bacterial translocation from gut to blood. Increased LPS levels (or LBP levels) have been reported in depressive patients (for e.g. PMID: 33488424). This should be discussed as this mechanism may trigger both depression and autoimmunity (https://www.sciencedirect.com/science/article/pii/S0092867416303981)

We thank the reviewer for this valuable suggestion and agree with this point. We tried to elucidate the relationship between intestinal barrier dysfunction, bacterial translocation and the immune system in depressed patients, taking into account the role of immune cells in this process. We have added the following excerpt:  
„It is also worth noting the increasing research on the impact of gut microflora and translocation of bacteria from the gut to the blood in patients with depressive disorders. The relationship between gut barrier dysfunction, bacterial translocation, and the immune system in depressed patients and the expression of activated monocytes in depressed patients has been established. Treg lymphocytes (which play a role in regulating the activation and effector activity of innate and adaptive immune response cells, as well as in controlling and suppressing the immune response against their own antigens) play an important role in this mechanism. There is an increased proportion of Treg cells in depressed patients. Moreover, there is a significant association between the gut microbiota and Treg. The reduction of Treg expansion may suppress the intestinal migration of these cells. Reduction of Treg of systemic origin may compromise the total number of intestinal Treg. Reduction of Treg in the intestinal mucosa may promote intestinal barrier damage and increased bacterial translocation found in some depressed patients. Furthermore, the presence of depression is associated with increased intestinal permeability or 'leaky gut' and increased bacterial translocation. There is also evidence that the microbiota affects the immune system and vice versa [80].

Depressed patients had significantly elevated levels of circulating I-FABP (intestinal fatty acid-binding protein, a marker of intestinal barrier function in peripheral blood) and LBP (lipopolysaccharide-binding protein). In addition, depressed patients with high LBP levels showed a significant reduction in circulating Treg compared to depressed patients with normal LBP. The balance of effector lymphoid cells and Treg cells may have a profound effect on how the intestinal mucosa responds to stressors that cause damage. Although there is no direct evidence for causality, this study supports the existence of a so-called gut-brain axis, linking gastrointestinal and immune system functions to the emotional and cognitive areas of the brain [80]”. (Pages 11-12)

4. “Cytokines act on the brain in two successive stages: the first is triggered by activation of primary afferent neurons innervating the area of the body where the inflammatory response is taking place. The second involves the slow diffusion of cytokines from the periventricular organs and choroid plexus to target brain parts, such as the amygdala complex. Although disease behavior is a normal host response to pathogens that are recognized by the innate immune system, there is evidence that the mechanisms that contribute to the development of disease behavior may play a role in the pathophysiology of depression [53]”.

5. Besides these remarks, I think that at least 1 explanatory scheme (2 or 3 even better) would add much value to this paper.

As suggested, we added 2 schemas: The first one summarizes potential pathomechanisms associated with the occurrence of depression in patients with AHT and the second presents the mechanisms of the relationship between depressive symptoms and selected drugs and their influence on the values of selected hormonal and inflammatory parameters.

Authors

Round 2

Reviewer 1 Report

The authors reacted in a good way to my criticism, thus I have no further comments.

Author Response

Thank you very much for your comments.

Reviewer 2 Report

The authors provided ad hoc modifications to the paper. Thank you.

I'd like to urge the authors to shorten the title of their article. It's way too long and not attractive enough in the present form

Author Response

Thank you very much for your comments.

This manuscript is a resubmission of an earlier submission. The following is a list of the peer review reports and author responses from that submission.